# Practical Quasi-Newton Methods for Training Deep Neural Networks

**Donald Goldfarb, Yi Ren, Achraf Bahamou**
Department of Industrial Engineering and Operations Research
Columbia University
New York, NY 10027
{goldfarb, yr2322, ab4689}@columbia.edu

## Abstract

We consider the development of practical stochastic quasi-Newton, and in particular Kronecker-factored block-diagonal BFGS and L-BFGS methods, for training deep neural networks (DNNs). In DNN training, the number of variables and components of the gradient $n$ is often of the order of tens of millions and the Hessian has $n^2$ elements. Consequently, computing and storing a full $n \times n$ BFGS approximation or storing a modest number of (step, change in gradient) vector pairs for use in an L-BFGS implementation is out of the question. In our proposed methods, we approximate the Hessian by a block-diagonal matrix and use the structure of the gradient and Hessian to further approximate these blocks, each of which corresponds to a layer, as the Kronecker product of two much smaller matrices. This is analogous to the approach in KFAC [30], which computes a Kronecker-factored block-diagonal approximation to the Fisher matrix in a stochastic natural gradient method. Because of the indefinite and highly variable nature of the Hessian in a DNN, we also propose a new damping approach to keep the upper as well as the lower bounds of the BFGS and L-BFGS approximations bounded. In tests on autoencoder feed-forward neural network models with either nine or thirteen layers applied to three datasets, our methods outperformed or performed comparably to KFAC and state-of-the-art first-order stochastic methods.

## 1   Introduction

We consider in this paper the development of practical stochastic quasi-Newton (QN), and in particular Kronecker-factored block-diagonal BFGS [6, 13, 16, 39] and L-BFGS [27], methods for training deep neural networks (DNNs). Recall that the BFGS method starts each iteration with a symmetric positive definite matrix $B$ (or $H = B^{-1}$) that approximates the current Hessian matrix (or its inverse), computes the gradient $\nabla \mathbf{f}$ of $f$ at the current iterate $\mathbf{x}$ and then takes a step $\mathbf{s} = -\alpha H \nabla \mathbf{f}$, where $\alpha$ is a step length (usually) determined by some inexact line-search procedure, such that $\mathbf{y}^\top \mathbf{s} > 0$, where $\mathbf{y} = \nabla \mathbf{f}^+ - \nabla \mathbf{f}$ and $\nabla \mathbf{f}^+$ is the gradient of $f$ at the new point $\mathbf{x}^+ = \mathbf{x} + \mathbf{s}$. The method then computes an updated approximation $B^+$ to $B$ (or $H^+$ to $H$) that remains symmetric and positive-definite and satisfies the so-called *quasi-Newton* (QN) condition $B^+ \mathbf{s} = \mathbf{y}$ (or equivalently, $H^+ \mathbf{y} = \mathbf{s}$). A consequence of this is that the matrix $B^+$ operates on the vector $\mathbf{s}$ in exactly the same way as the average of the Hessian matrix along the line segment between $\mathbf{x}$ and $\mathbf{x}^+$ operates on $\mathbf{s}$.

In DNN training, the number of variables and components of the gradient $n$ is often of the order of tens of millions and the Hessian has $n^2$ elements. Hence, computing and storing a full $n \times n$ BFGS approximation or storing $p$ $(\mathbf{s}, \mathbf{y})$ pairs, where $p$ is approximately $10$ or larger for use in an L-BFGS implementation, is out of the question. Consequently, in our methods, we approximate the Hessian by a block-diagonal matrix, where each diagonal block corresponds to a layer, further approximating them as the Kronecker product of two much smaller matrices, as in [30, 5, 19, 10].

**Literature Review on Using Second-order Information for DNN Training.** For solving the stochastic optimization problems with high-dimensional data that arise in machine learning (ML), stochastic gradient descent (SGD) [36] and its variants are the methods that are most often used, especially for training DNNs. These variants include such methods as AdaGrad [12], RMSprop [21], and Adam [24], all of which scale the stochastic gradient by a diagonal matrix based on estimates of the first and second moments of the individual gradient components. Nonetheless, there has been a lot of effort to find ways to take advantage of second-order information in solving ML optimization problems. Approaches have run the gamut from the use of a diagonal re-scaling of the stochastic gradient, based on the secant condition associated with quasi-Newton (QN) methods [4], to sub-sampled Newton methods (e.g. see [43], and references therein), including those that solve the Newton system using the linear conjugate gradient method (see [8]).

In between these two extremes are stochastic methods that are based either on QN methods or generalized Gauss-Newton (GGN) and natural gradient [1] methods. For example, a stochastic L-BFGS method for solving strongly convex problems was proposed in [9] that uses sampled Hessian-vector products rather than gradient differences, which was proved in [33] to be linearly convergent by incorporating the variance reduction technique (SVRG [23]) to alleviate the effect of noisy gradients. A closely related variance reduced block L-BFGS method was proposed in [17]. A regularized stochastic BFGS method was proposed in [31], and an online L-BFGS method was proposed in [32] for strongly convex problems and extended in [28] to incorporate SVRG variance reduction. Stochastic BFGS and L-BFGS methods were also developed for online convex optimization in [38]. For nonconvex problems, a damped L-BFGS method which incorporated SVRG variance reduction was developed and its convergence properties was studied in [41].

GGN methods that approximate the Hessian have been proposed, including the Hessian-free method [29] and the Krylov subspace method [40]. Variants of the closely related natural gradient method that use block-diagonal approximations to the Fisher information matrix, where blocks correspond to layers, have been proposed in e.g. [20, 11, 30, 14]. Using further approximation of each of these (empirical) Fisher matrix and GGN blocks by the Kronecker product of two much smaller matrices, the efficient KFAC [30], KFRA [5], EKFAC [15], and Shampoo [19] methods were developed. See also [2] and [10], [37], which combine both Hessian and covariance (Fisher-like) matrix information in stochastic Newton type methods, Also, methods are given in [26, 42] that replace the Kullback-Leibler divergence by the Wasserstein distance to define the natural gradient, but with a greater computational cost.

**Our Contributions.** The main contributions of this paper can be summarized as follows:

1. New BFGS and limited-memory variants (i.e. L-BFGS) that take advantage of the structure of feed-forward DNN training problems;
2. Efficient non-diagonal second-order algorithms for deep learning that require a comparable amount of memory and computational cost per iteration as first-order methods;
3. A new damping scheme for BFGS and L-BFGS updating of an inverse Hessian approximation, that not only preserves its positive definiteness, but also limits the decrease (and increase) in its smallest (and largest) eigenvalues for non-convex problems;
4. A novel application of Hessian-action BFGS;
5. The first proof of convergence (to the best of our knowledge) of a stochastic Kronecker-factored quasi-Newton method.

## 2  Kronecker-factored Quasi-Newton Method for DNN

After reviewing the computations used in DNN training, we describe the Kronecker structures of the gradient and Hessian for a single data point, followed by their extension to approximate expectations of these quantities for multiple data-points and give a generic algorithm that employs BFGS (or L-BFGS) approximations for the Hessians.

**Deep Neural Networks.** We consider a feed-forward DNN with $L$ layers, defined by weight matrices $W_l$ (whose last columns are bias vectors $b_l$), activation functions $\phi_l$ for $l \in \{1 \ldots L\}$ and loss function $\mathcal{L}$. For a data-point $(x, y)$, the loss $\mathcal{L}(a_L, y)$ between the output $a_L$ of the DNN and $y$ is a non-convex function of $\theta = \left[ \text{vec}(W_1)^\top, \ldots, \text{vec}(W_L)^\top \right]^\top$. The network's forward and backward pass for a single input data point $(x, y)$ is described in Algorithm 1.

---

**Algorithm 1** Forward and backward pass of DNN for a single data-point

---

1: Given input $(x, y)$, weights (and biases) $W_l$, and activations $\phi_l$ for $l \in [1, L]$
2: $\mathbf{a}_0 = x$; **for** $l = 1, .., L$ **do** $\bar{\mathbf{a}}_{l-1} = (\mathbf{a}_{l-1}, 1)$; $\mathbf{h}_l = W_l \bar{\mathbf{a}}_{l-1}$; $\mathbf{a}_l = \phi_l(\mathbf{h}_l)$
3: $\mathcal{D}\mathbf{a}_L \leftarrow \left. \frac{\partial \mathcal{L}(z,y)}{\partial z} \right|_{z=\mathbf{a}_L}$
4: **for** $l = L, .., 1$ **do** $\mathbf{g}_l = \mathcal{D}\mathbf{a}_l \odot \phi_l'(\mathbf{h}_l)$; $\mathcal{D}W_l = \mathbf{g}_i \bar{\mathbf{a}}_{i-1}^\top$; $\mathcal{D}\mathbf{a}_{l-1} = W_l^\top \mathbf{g}_l$

---

For a training dataset that contains multiple data-points indexed by $i = 1, ..., I$, let $f(i; \theta)$ denote the loss for the $i$th data-point. Then, viewing the dataset as an empirical distribution, the total loss function $f(\theta)$ that we wish to minimize is

$$f(\theta) := \mathbb{E}_i[f(i; \theta)] := \frac{1}{I} \sum_{i=1}^{I} f(i; \theta).$$

**Single Data-point: Layer-wise Structure of the Gradient and Hessian.** Let $\nabla \mathbf{f}_l$ and $\nabla^2 f_l$ denote, respectively, the restriction of $\nabla \mathbf{f}$ and $\nabla^2 f$ to the weights $W_l$ in layer $l = 1, \ldots, L$. For a single data-point $\nabla \mathbf{f}_l$ and $\nabla^2 f_l$ have a tensor (Kronecker) structure, as shown in [30] and [5]. Specifically,

$$\nabla \mathbf{f}_l(i) = \mathbf{g}_l(i)(\mathbf{a}_{l-1}(i))^\top, \quad \text{equivalently,} \quad \text{vec}(\nabla \mathbf{f}_l(i)) = \mathbf{a}_{l-1}(i) \otimes \mathbf{g}_l(i), \tag{1}$$

$$\nabla^2 f_l(i) = (\mathbf{a}_{l-1}(i)(\mathbf{a}_{l-1}(i))^\top) \otimes G_l(i), \tag{2}$$

where the pre-activation gradient $\mathbf{g}_l(i) = \frac{\partial f(i)}{\partial \mathbf{h}_l(i)}$, and the pre-activation Hessian $G_l(i) = \frac{\partial^2 f(i)}{\partial \mathbf{h}_l(i)^2}$. Our algorithm uses an approximation to $(G_l(i))^{-1}$, which is updated via the BFGS updating formulas based upon a secant condition that relates the change in $\mathbf{g}_l(i)$ with the change in $\mathbf{h}_l(i)$.

Although we focus on fully-connected layers in this paper, the idea of Kronecker-factored approximations to the diagonal blocks $\nabla^2 f_l$, $l = 1, \ldots, L$ of the Hessian can be extended to other layers used in deep learning, such as convolutional and recurrent layers.

**Multiple Data-points: Kronecker-factored QN Approach.** Now consider the case where we have a dataset of $I$ data-points indexed by $i = 1, \ldots, I$. By (2), we have

$$\mathbb{E}_i[\nabla^2 f_l(i)] \approx \mathbb{E}_i \left[ \mathbf{a}_{l-1}(i)(\mathbf{a}_{l-1}(i))^\top \right] \otimes \mathbb{E}_i \left[ G_l(i) \right] := A_l \otimes G_l \tag{3}$$

Note that the approximation in (3) that the expectation of the Kronecker product of two matrices equals the Kronecker product of their expectations is the same as the one used by KFAC [30]. Now, based on this structural approximation, we use $H^l = H_a^l \otimes H_g^l$ as our QN approximation to $\left( \mathbb{E}_i[\nabla^2 f_l(i)] \right)^{-1}$, where $H_a^l$ and $H_g^l$ are positive definite approximations to $A_l^{-1}$ and $G_l^{-1}$, respectively. Hence, using our layer-wise block-diagonal approximation to the Hessian, a step in our algorithm for each layer $l$ is computed as

$$\text{vec}(W_l^+) - \text{vec}(W_l) = -\alpha H^l \text{vec}\left( \widehat{\nabla \mathbf{f}_l} \right) = -\alpha (H_a^l \otimes H_g^l) \text{vec}\left( \widehat{\nabla \mathbf{f}_l} \right) = -\alpha \text{vec}\left( H_g^l \widehat{\nabla \mathbf{f}_l} H_a^l \right), \tag{4}$$

where $\widehat{\nabla \mathbf{f}_l}$ denotes the estimate to $\mathbb{E}_i[\nabla \mathbf{f}_l(i)]$ and $\alpha$ is the learning rate. After computing $W_l^+$ and performing another forward/backward pass, our method computes or updates $H_a^l$ and $H_g^l$ as follows:

1. For $H_g^l$, we use a damped version of BFGS (or L-BFGS) (See Section 3) based on the $(\mathbf{s}, \mathbf{y})$ pairs corresponding to the average change in $\mathbf{h}_l(i)$ and in the gradient with respect to $\mathbf{h}_l(i)$; i.e.,

$$\mathbf{s}_g^l = \mathbb{E}_i[\mathbf{h}_l^+(i)] - \mathbb{E}_i[\mathbf{h}_l(i)], \qquad \mathbf{y}_g^l = \mathbb{E}_i[\mathbf{g}_l^+(i)] - \mathbb{E}_i[\mathbf{g}_l(i)]. \tag{5}$$

2. For $H_a^l$ we use the "Hessian-action" BFGS method described in Section 4. The issue of possible singularity of the positive semi-definite matrix $A_l$ approximated by $(H_a^l)^{-1}$ is also addressed there by incorporating a **Levenberg-Marquardt (LM)** damping term.

Algorithm 2 gives a high-level summary of our **K-BFGS** or **K-BFGS(L)** algorithms (which use BFGS or L-BFGS to update $H_g^l$, respectively). See Algoirthm 4 in the appendix for a detailed pseudocode. The use of mini-batches is described in Section 6. Note that, an additional forward-backward pass is used in Algorithm 2 because the quantities in (5) need to be estimated using the same mini-batch.

---

**Algorithm 2** High-level summary of K-BFGS / K-BFGS(L)

---

**Require:** Given initial weights $\theta$, batch size $m$, learning rate $\alpha$
1: **for** $k = 1, 2, \ldots$ **do**
2:      Sample mini-batch of size $m$: $M_k = \{\xi_{k,i}, i = 1, \ldots, m\}$
3:      Perform a forward-backward pass over the current mini-batch $M_k$ (see Algorithm 1)
4:      **for** $l = 1, \ldots, L$ **do** $p_l = H_g^l \widehat{\nabla} \mathbf{f}_l H_a^l$; $W_l = W_l - \alpha \cdot p_l$
5:      Perform another forward-backward pass over $M_k$ to get $(\mathbf{s}_g^l, \mathbf{y}_g^l)$
6:      Use damped BFGS or L-BFGS to update $H_g^l$ ($l = 1, ..., L$) (see Section 3, in particular Algorithm 3)
7:      Use Hessian-action BFGS to update $H_a^l$ ($l = 1, ..., L$) (see Section 4 )

---

# 3    BFGS and L-BFGS for $G_l$

**Damped BFGS Updating.** It is well-known that training a DNN is a non-convex optimization problem. As (2) and (3) show, this non-convexity manifests in the fact that $G_l \succ 0$ often does not hold. Thus, for the BFGS update of $H_g^l$, the approximation to $G_l^{-1}$, to remain positive definite, we have to ensure that $(\mathbf{s}_g^l)^\top \mathbf{y}_g^l > 0$. Due to the stochastic setting, ensuring this condition by line-search, as is done in deterministic settings, is impractical. In addition, due to the large changes in curvature in DNN models that occur as the parameters are varied, we also need to suppress large changes to $H_g^l$ as it is updated. To deal with both of these issues, we propose a **double damping (DD)** procedure (Algorithm 3), which is based upon **Powell's damped-BFGS** approach [35], for modifying the $(\mathbf{s}_g^l, \mathbf{y}_g^l)$ pair. To motivate Algorithm 3, consider the formulas used for BFGS updating of $B$ and $H$:

$$B^+ = B - \frac{B\mathbf{s}\mathbf{s}^\top B}{\mathbf{s}^\top B\mathbf{s}} + \rho \mathbf{y}\mathbf{y}^\top, \quad H^+ = (I - \rho\mathbf{s}\mathbf{y}^\top)H(I - \rho\mathbf{y}\mathbf{s}^\top) + \rho\mathbf{s}\mathbf{s}^\top, \tag{6}$$

where $\rho = \frac{1}{\mathbf{s}^\top \mathbf{y}} > 0$. If we can ensure that $0 < \frac{\mathbf{y}^\top H\mathbf{y}}{\mathbf{s}^\top \mathbf{y}} \leq \frac{1}{\mu_1}$ and $0 < \frac{\mathbf{s}^\top \mathbf{s}}{\mathbf{s}^\top \mathbf{y}} \leq \frac{1}{\mu_2}$ , then we can obtain the following bounds:

$$\|B^+\| \leq \|B - \frac{B\mathbf{s}\mathbf{s}^\top B}{\mathbf{s}^\top B\mathbf{s}}\| + \|\rho\mathbf{y}\mathbf{y}^\top\| \leq \|B\| + \|\frac{B^{1/2}H^{1/2}\mathbf{y}\mathbf{y}^\top H^{1/2}B^{1/2}}{\mathbf{s}^\top \mathbf{y}}\| \tag{7}$$

$$\leq \|B\| + \|B\|\frac{\|H^{1/2}\mathbf{y}\|^2}{\mathbf{s}^\top \mathbf{y}} \leq \|B\|\left(1 + \frac{\mathbf{y}^\top H\mathbf{y}}{\mathbf{s}^\top \mathbf{y}}\right) \leq \|B\|\left(1 + \frac{1}{\mu_1}\right) \tag{8}$$

and

$$\|H^+\| \leq \|H^{1/2} - \frac{\mathbf{s}\mathbf{y}^\top H^{1/2}}{\mathbf{s}^\top \mathbf{y}}\|^2 + \|\frac{\mathbf{s}\mathbf{s}^\top}{\mathbf{s}^\top \mathbf{y}}\| \leq \left(\|H^{1/2}\| + \frac{\|\mathbf{s}\|\|H^{1/2}\mathbf{y}\|}{\mathbf{s}^\top \mathbf{y}}\right)^2 + \frac{\|\mathbf{s}\|^2}{\mathbf{s}^\top \mathbf{y}} \tag{9}$$

$$\leq \left(\|H^{1/2}\| + (\frac{\mathbf{s}^\top \mathbf{s}}{\mathbf{s}^\top \mathbf{y}})^{1/2}(\frac{\mathbf{y}^\top H\mathbf{y}}{\mathbf{s}^\top \mathbf{y}})^{1/2}\right)^2 + \frac{\mathbf{s}^\top \mathbf{s}}{\mathbf{s}^\top \mathbf{y}} \leq \left(\|H^{1/2}\| + \frac{1}{\sqrt{\mu_1\mu_2}}\right)^2 + \frac{1}{\mu_2}. \tag{10}$$

Thus, the change in $B$ (and $H$) is controlled if $\frac{\mathbf{y}^\top H\mathbf{y}}{\mathbf{s}^\top \mathbf{y}} \leq \frac{1}{\mu_1}$ and $\frac{\mathbf{s}^\top \mathbf{s}}{\mathbf{s}^\top \mathbf{y}} \leq \frac{1}{\mu_2}$. Our DD approach is a two-step procedure, where the first step (i.e. Powell's damping of $H$) guarantees that $\frac{\mathbf{y}^\top H\mathbf{y}}{\mathbf{s}^\top \mathbf{y}} \leq \frac{1}{\mu_1}$ and the second step (i.e., Powell's damping with $B = I$) guarantees that $\frac{\mathbf{s}^\top \mathbf{s}}{\mathbf{s}^\top \mathbf{y}} \leq \frac{1}{\mu_2}$. Note that there is no guarantee of $\frac{\mathbf{y}^\top H\mathbf{y}}{\mathbf{s}^\top \mathbf{y}} \leq \frac{1}{\mu_1}$ after the second step. However, we can skip updating $H$ in this case so that the bounds on these matrices hold. In our implementation, we always do the update, since in empirical testing, we observed that at least 90% of the pairs satisfy $\frac{\mathbf{y}^\top H\mathbf{y}}{\mathbf{s}^\top \mathbf{y}} \leq \frac{2}{\mu_1}$. See Section C in the appendix for more details on damping.

---
**Algorithm 3** Double Damping (DD)
---
1: **Input: s, y; Output: $\tilde{\mathbf{s}}, \tilde{\mathbf{y}}$; Given:** $H, \mu_1, \mu_2$
2: **if** $\mathbf{s}^\top \mathbf{y} < \mu_1 \mathbf{y}^\top H \mathbf{y}$ **then** $\theta_1 = \frac{(1-\mu_1)\mathbf{y}^\top H \mathbf{y}}{\mathbf{y}^\top H \mathbf{y} - \mathbf{s}^\top \mathbf{y}}$ **else** $\theta_1 = 1$
3: $\tilde{\mathbf{s}} = \theta_1 \mathbf{s} + (1 - \theta_1)H\mathbf{y}$ {Powell's damping on $H$}
4: **if** $\tilde{\mathbf{s}}^\top \mathbf{y} < \mu_2 \tilde{\mathbf{s}}^\top \tilde{\mathbf{s}}$ **then** $\theta_2 = \frac{(1-\mu_2)\tilde{\mathbf{s}}^\top \tilde{\mathbf{s}}}{\tilde{\mathbf{s}}^\top \tilde{\mathbf{s}} - \tilde{\mathbf{s}}^\top \mathbf{y}}$ **else** $\theta_2 = 1$
5: $\tilde{\mathbf{y}} = \theta_2 \mathbf{y} + (1 - \theta_2)\tilde{\mathbf{s}}$ {Powell's damping with $B = I$}
6: **return** $\tilde{s}, \tilde{y}$
---

**L-BFGS Implementation.** L-BFGS can also be used to update $H_g^l$. However, implementing L-BFGS using the standard "two-loop recursion" (see Algorithm 7.4 in [34]) is not efficient. This is because the main work in computing $H_g^l \widehat{\nabla \mathbf{f}_l} H_a^l$ in line 4 of Algorithm 2 would require $4p$ matrix-vector multiplications, each requiring $O(d_i d_o)$ operations, where $p$ denotes the number of $(\mathbf{s}, \mathbf{y})$ pairs stored by L-BFGS. (Recall that $\widehat{\nabla \mathbf{f}_l} \in R^{d_o \times d_i}$.) Instead, we use a "non-loop" implementation [7] of L-BFGS, whose main work involves 2 matrix-matrix multiplications, each requiring $O(pd_i d_o)$ operations. When $p$ is not small (we used $p = 100$ in our tests), and $d_i$ and $d_o$ are large, this is much more efficient, especially on GPUs.

## 4 "Hessian action" BFGS for $A_l$

In addition to approximating $G_l^{-1}$ by $H_g^l$ using BFGS, we also propose approximating $A_l^{-1}$ by $H_a^l$ using BFGS. Note that $A_l$ does not correspond to some Hessian of the objective function. However, we can generate $(\mathbf{s}, \mathbf{y})$ pairs for it by "Hessian action" (see e.g. [9, 17, 18]).

**Connection between Hessian-action BFGS and Matrix Inversion.** In our methods, we choose $\mathbf{s} = H_a^l \cdot \mathbb{E}_i[\mathbf{a}_{l-1}(i)]$ and $\mathbf{y} = A_l\mathbf{s}$, which as we now show, is closely connected to using the Sherman-Morrison modification formula to invert $A_l$. In particular, suppose that $A^+ = A + c \cdot \mathbf{aa}^\top$; i.e., only a rank-one update is made to $A$. This corresponds to the case where the information of $A$ is accumulated from iteration to iteration, and the size of the mini-batch is 1 or $\mathbf{a}$ represents the average of the vectors $\mathbf{a}(i)$ from multiple data-points.

**Theorem 1.** *Suppose that $A$ and $H$ are symmetric and positive definite, and that $H = A^{-1}$. If we choose $\mathbf{s} = H\mathbf{a}$ and $\mathbf{y} = A^+\mathbf{s}$, where $A^+ = A + c \cdot \mathbf{aa}^\top$ ($c > 0$). Then, the $H^+$ generated by any QN update in the **Broyden family***

$$H^+ = H - \sigma H\mathbf{yy}^\top H + \rho\mathbf{ss}^\top + \phi(\mathbf{y}^\top H\mathbf{y})\mathbf{hh}^\top, \tag{11}$$

*where $\rho = 1/\mathbf{s}^\top \mathbf{y}$, $\sigma = 1/\mathbf{y}^\top H\mathbf{y}$, $\mathbf{h} = \rho\mathbf{s} - \sigma H\mathbf{y}$ and $\phi$ is a scalar parameter in $[0, 1]$, equals $(A^+)^{-1}$. Note that $\phi = 1$ yields the BFGS update (6) and $\phi = 0$ yields the DFP update.*

*Proof.* If $\mathbf{s} = H\mathbf{a}$ and $\mathbf{y} = A^+\mathbf{s}$, then $\mathbf{h} = 0$, so all choices of $\phi$ yield the same matrix $H^+$. Since $H^+A^+\mathbf{s} = H^+\mathbf{y} = \mathbf{s}$ and for any vector $\mathbf{v}$ that is orthogonal to $\mathbf{a}$, $H^+A^+\mathbf{v} = H^+A\mathbf{v} = \mathbf{v}$, since $\mathbf{s}^\top A\mathbf{v} = 0$ and $\mathbf{y}^\top HA\mathbf{v} = 0$, it follows that $H^+A^+ = I$, using the fact that $\mathbf{s}$ together with any linearly independent set of $n - 1$ vectors orthogonal to $\mathbf{a}$ spans $R^n$. (Note that $\mathbf{s}^\top \mathbf{a} = \mathbf{a}^\top H\mathbf{a} > 0$, since $H \succ 0 \Rightarrow$ that $\mathbf{s}$ is not orthogonal to $\mathbf{a}$.) $\qquad\square$

In fact, all updates in the Broyden family are equivalent to applying the Sherman-Morrison modification formula to $A^+ = A + c \cdot \mathbf{aa}^\top$, given $H = A^{-1}$, since after substituting for $\mathbf{s}$ and $\mathbf{y}$ in (11) and simplifying, one obtains

$$H^+ = H - H\mathbf{a}(c^{-1} + \mathbf{a}^\top H\mathbf{a})^{-1}\mathbf{a}^\top H.$$

When using momentum, $A^+ = \beta A + (1 - \beta)\mathbf{aa}^\top$ ($0 < \beta < 1$). Hence, if we still want Theorem 1 to hold, we have to scale $H$ by $1/\beta$ before updating it. This, however, turns out to be unstable. Hence, in practice, we use the non-scaled version of "Hessian action" BFGS.

**Levenberg-Marquardt Damping for $A_l$.** Since $A_l = \mathbb{E}_i \left[ (\mathbf{a}_{l-1}(i)(\mathbf{a}_{l-1}(i))^\top) \right] \succeq 0$ may not be positive definite, or may have very small positive eigenvalues, we add an **Levenberg-Marquardt (LM) damping** term to make our "Hessian-action" BFGS stable; i.e., we use $A_l + \lambda_A I_A$ instead of $A_l$, when we update $H_a^l$. Specifically, "Hessian action" BFGS for $A_l$ is performed as

1. $A_l = \beta \cdot A_l + (1 - \beta) \cdot \mathbb{E}_i \left[ \mathbf{a}_{l-1}(i) \mathbf{a}_{l-1}(i)^\top \right]$; $A_l^{\mathrm{LM}} = A_l + \lambda_A I_A$.
2. $\mathbf{s}_a^l = H_a^l \cdot \mathbb{E}_i[\mathbf{a}_{l-1}(i)]$, $\mathbf{y}_a^l = A_l^{\mathrm{LM}} \mathbf{s}_a^l$; use BFGS with $(\mathbf{s}_a^l, \mathbf{y}_a^l)$ to update $H_a^l$.

# 5  Convergence Analysis

Following the framework for stochastic quasi-Newton methods (SQN) established in [41] for solving nonconvex stochastic optimization problems (see Section B in the appendix for this framework), we prove that, under fairly standard assumptions, for our K-BFGS(L) algorithm with skipping DD and exact inversion on $A_l$ (see Algorithm 5 in Section B), the number of iterations $N$ needed to obtain $\frac{1}{N} \sum_{k=1}^N \mathbb{E}[\|\nabla \mathbf{f}(\theta_k)\|^2] \leq \epsilon$ is $N = O(\epsilon^{-\frac{1}{1-\beta}})$, for step size $\alpha_k$ chosen proportional to $k^{-\beta}$, where $\beta \in (0.5, 1)$ is a constant. Our proofs, which are delayed until Section B, make use of the following assumptions, the first two of which, were made in [41].

**AS. 1.** $f : \mathbb{R}^n \to \mathbb{R}$ is continuously differentiable. $f(\theta) \geq f^{low} > -\infty$, for any $\theta \in \mathbb{R}^n$. $\nabla \mathbf{f}$ is globally L-Lipschitz continuous; namely for any $x, y \in \mathbb{R}^n$, $\|\nabla \mathbf{f}(x) - \nabla \mathbf{f}(y)\| \leq L\|x - y\|$.

**AS. 2.** For any iteration $k$, the stochastic gradient $\widehat{\nabla \mathbf{f}}_k = \widehat{\nabla \mathbf{f}}(\theta_k, \xi_k)$ satisfies:

a) $\mathbb{E}_{\xi_k} \left[ \widehat{\nabla \mathbf{f}}(\theta_k, \xi_k) \right] = \nabla \mathbf{f}(\theta_k)$, b) $\mathbb{E}_{\xi_k} \left[ \left\| \widehat{\nabla \mathbf{f}}(\theta_k, \xi_k) - \nabla \mathbf{f}(\theta_k) \right\|^2 \right] \leq \sigma^2$, where $\sigma > 0$, and

$\xi_k, k = 1, 2, \ldots$ are independent samples that are independent of $\{\theta_j\}_{j=1}^k$.

**AS. 3.** The activation functions $\phi_l$ have bounded values: $\exists \varphi > 0$ s.t. $\forall l, \forall h, |\phi_l(h)| \leq \varphi$.

To use the convergence analysis in [41], we need to show that the block-diagonal approximation of the inverse Hessian used in Algorithm 5 satisfies the assumption that it is bounded above and below by positive-definite matrices. Given the Kronecker structure of our Hessian inverse approximation, it suffices to prove boundness of both $H_a^l(k)$ and $H_g^l(k)$ for all iterations $k$. Making the additional assumption AS.3, we are able to prove Lemma 1, and hence Lemma 3, below. Note that many popular activation functions satisfy AS.3, such as sigmoid and tanh.

**Lemma 1.** Suppose that AS.3 holds. There exist two positive constants $\underline{\kappa}_a$, $\bar{\kappa}_a$ such that $\underline{\kappa}_a I \preceq H_a^l(k) \preceq \bar{\kappa}_a I, \forall k, l$.

**Lemma 2.** There exist two positive constants $\underline{\kappa}_g$ and $\bar{\kappa}_g$, such that $\underline{\kappa}_g I \preceq H_g^l(k) \preceq \bar{\kappa}_g I, \forall k, l$.

**Lemma 3.** Suppose that AS.3 holds. Let $\theta_{k+1} = \theta_k - \alpha_k H_k \widehat{\nabla \mathbf{f}}_k$ be the step taken in Algorithm 5. There exists two positive constants $\underline{\kappa}$, $\bar{\kappa}$ such that $\underline{\kappa} I \preceq H_k \preceq \bar{\kappa} I, \forall k$.

Using Lemma 3, we can now apply Theorem 2.8 in [41] to prove the convergence of Algorithm 5:

**Theorem 2.** Suppose that assumptions AS.1-3 hold for $\{\theta_k\}$ generated by Algorithm 5 with mini-batch size $m_k = m$ for all $k$, and $\alpha_k$ is chosen as $\alpha_k = \frac{\underline{\kappa}}{L\bar{\kappa}^2} k^{-\beta}$, with $\beta \in (0.5, 1)$. Then

$$\frac{1}{N} \sum_{k=1}^N \mathbb{E} \left[ \|\nabla \mathbf{f}(\theta_k)\|^2 \right] \leq \frac{2L \left( M_f - f^{low} \right) \bar{\kappa}^2}{\underline{\kappa}^2} N^{\beta-1} + \frac{\sigma^2}{(1-\beta)m} \left( N^{-\beta} - N^{-1} \right)$$

where $N$ denotes the iteration number and $M_f > 0$ depends only on $f$. Moreover, for a given $\epsilon \in (0, 1)$, to guarantee that $\frac{1}{N} \sum_{k=1}^N \mathbb{E} \left[ \|\nabla \mathbf{f}(\theta_k)\|^2 \right] < \epsilon$, the number of iterations $N$ needed is at most $O \left( \epsilon^{-\frac{1}{1-\beta}} \right)$.

Note: other theorems in [41], namely Theorems 2.5 and 2.6, also apply here under our assumptions.

# 6  Experiments

Before we present some experimental results, we address the use of moving averages, and the computational and storage requirements of the algorithms that we tested.

**Mini-batch and Moving Average.** Clearly, using the whole dataset at each iteration is inefficient; hence, we use a mini-batch to estimate desired quantities. We use $\overline{X}$ to denote the averaged value of $X$ across the mini-batch for any quantity $X$. To incorporate information from the past as well as reducing the variability, we use an exponentially decaying moving average to estimate desired quantities with decay parameter $\beta \in (0, 1)$:

1. To estimate the gradient $\mathbb{E}_i[\nabla \mathbf{f}(i)]$, at each iteration, we update $\widehat{\nabla \mathbf{f}} = \beta \cdot \widehat{\nabla \mathbf{f}} + (1 - \beta) \cdot \overline{\nabla \mathbf{f}}$.

2. $H_a^l$: To estimate $A_l$, at each iteration we update $\widehat{A_l} = \beta \cdot \widehat{A_l} + (1 - \beta) \cdot \overline{\mathbf{a}_{l-1}\mathbf{a}_{l-1}^\top}$. Note that although we compute $\mathbf{s}_a^l$ as $H_a^l \cdot \overline{\mathbf{a}_{l-1}}$, we update $\widehat{A_l}$ with $\overline{\mathbf{a}_{l-1}\mathbf{a}_{l-1}^\top}$ (i.e. the average $\mathbf{a}_{l-1}(i)\mathbf{a}_{l-1}(i)^\top$ over the minibatch, not $\overline{\mathbf{a}_{l-1}} \cdot (\overline{\mathbf{a}_{l-1}})^\top$).

3. $H_g^l$: BFGS "uses" momentum implicitly incorporated in the matrices $H_g^l$. To further stabilize the BFGS update, we also use a moving-averaged $(\mathbf{s}_g^l, \mathbf{y}_g^l)$ (before damping); i.e., We update $\mathbf{s}_g^l = \beta \cdot \mathbf{s}_g^l + (1 - \beta) \cdot \left( \overline{\mathbf{h}_l^+} - \overline{\mathbf{h}_l} \right)$, and $\mathbf{y}_g^l = \beta \cdot \mathbf{y}_g^l + (1 - \beta) \cdot \left( \overline{\mathbf{g}_l^+} - \overline{\mathbf{g}_l} \right)$.

Finally, when computing $\overline{\mathbf{h}_l^+}$ and $\overline{\mathbf{g}_l^+}$, we use the same mini-batch as was used to compute $\overline{\mathbf{h}_l}$ and $\overline{\mathbf{g}_l}$. This doubles the number of forward-backward passes at each iteration.

**Storage and Computational Complexity.** Tables 1 and 2 compare the storage and computational requirements, respectively, for a layer with $d_i$ inputs and $d_o$ outputs for K-BFGS, K-BFGS(L), KFAC, and Adam/RMSprop. We denote the size of mini-batch by $m$, the number of $(\mathbf{s}, \mathbf{y})$ pairs stored for L-BFGS by $p$, and the frequency of matrix inversion in KFAC by $T$. Besides the requirements listed in Table 1, all algorithms need storage for the parameters $W_l$ and the estimate of the gradient, $\widehat{\nabla \mathbf{f}_1}$, (i.e. $O(d_i d_o)$). Besides the work listed in Table 2, all algorithms also need to do a forward-backward pass to compute $\nabla \mathbf{f}_l$ as well as updating $W_l$, (i.e. $O(m d_i d_o)$). Also note that, even though we use big-$O$ notation in these tables, the constants for all of the terms in each of the rows are roughly at the same level and relatively small.

In Table 2, for K-BFGS and K-BFGS(L), "Additional pass" refers to Line 5 of Algorithm 2; under "Curvature", $O(m d_i^2)$ arises from "Hessian action" BFGS to update $H_a^l$ (see the algorithm at the end of Section 4), $O(m d_o)$ arises from (5), $O(d_o^2)$ arises from updating $H_g^l$ (only for K-BFGS); and "Step $\Delta W_l$" refers to (4). For KFAC, referring to Algorithm 7 (in the appendix), "Additional pass" refers to Line 7; under "Curvature", $O(m d_i^2 + m d_o^2)$ refers to Line 8, and $O(\frac{1}{T}d_i^3 + \frac{1}{T}d_o^3)$ refers to Line 10; and "Step $\Delta W_l$" refers to Line 5.

From Table 1, we see that the Kronecker property enables K-BFGS and K-BFGS(L) (as well as KFAC) to have storage requirements comparable to those of first-order methods. Moreover, from Table 2, we see that K-BFGS and K-BFGS(L) require less computation per iteration than KFAC, since they only involve matrix multiplications, whereas KFAC requires matrix inversions which depend cubically on both $d_i$ and $d_o$. The cost of matrix inversion in KFAC (and singular value decomposition in [19]) is amortized by performing these operations only once every $T$ iterations; nonetheless, these amortized operations usually become much slower than matrix multiplication as models scale up.

Table 1: Storage

| Algorithm | $\nabla f_l \odot \nabla f_l$ | $A$ | $G$ | Total |
|---|---|---|---|---|
| K-BFGS | — | $O(d_i^2)$ | $O(d_o^2)$ | $O(d_i^2 + d_o^2 + d_i d_o)$ |
| K-BFGS(L) | — | $O(d_i^2)$ | $O(p d_o)$ | $O(d_i^2 + d_i d_o + p d_o)$ |
| KFAC | — | $O(d_i^2)$ | $O(d_o^2)$ | $O(d_i^2 + d_o^2 + d_i d_o)$ |
| Adam/RMSprop | $O(d_i d_o)$ | — | — | $O(d_i d_o)$ |

Table 2: Computation per iteration

| Algorithm | Additional pass | Curvature | Step $\Delta W_l$ |
|---|---|---|---|
| K-BFGS | $O(m d_i d_o)$ | $O(m d_i^2 + m d_o + d_o^2)$ | $O(d_i^2 d_o + d_o^2 d_i)$ |
| K-BFGS(L) | $O(m d_i d_o)$ | $O(m d_i^2 + m d_o)$ | $O(d_i^2 d_o + p d_i d_o)$ |
| KFAC | $O(m d_i d_o)$ | $O(m d_i^2 + m d_o^2 + \frac{1}{T}d_i^3 + \frac{1}{T}d_o^3)$ | $O(d_i^2 d_o + d_o^2 d_i)$ |
| Adam/RMSprop | — | $O(d_i d_o)$ | $O(d_i d_o)$ |

**Experimental Results.** We tested K-BFGS and K-BFGS(L), as well as KFAC, Adam/RMSprop and SGD-m (SGD with momentum) on three autoencoder problems, namely, MNIST [25], FACES, and CURVES, which are used in e.g. [22, 29, 30], except that we replaced the sigmoid activation with ReLU. See Section D in the appendix for a complete description of these problems and the competing algorithms.

Since one can view Powell's damping with $B = I$ as LM damping, we write $\mu_2 = \lambda_G$, where $\lambda_G$ denotes the LM damping parameter for $G_l$. We then define $\lambda = \lambda_A \lambda_G$ as the overall damping term of our QN approximation. To simplify matters, we chose $\lambda_A = \lambda_G = \sqrt{\lambda}$, so that we needed to tune only one hyper-parameter (HP) $\lambda$.

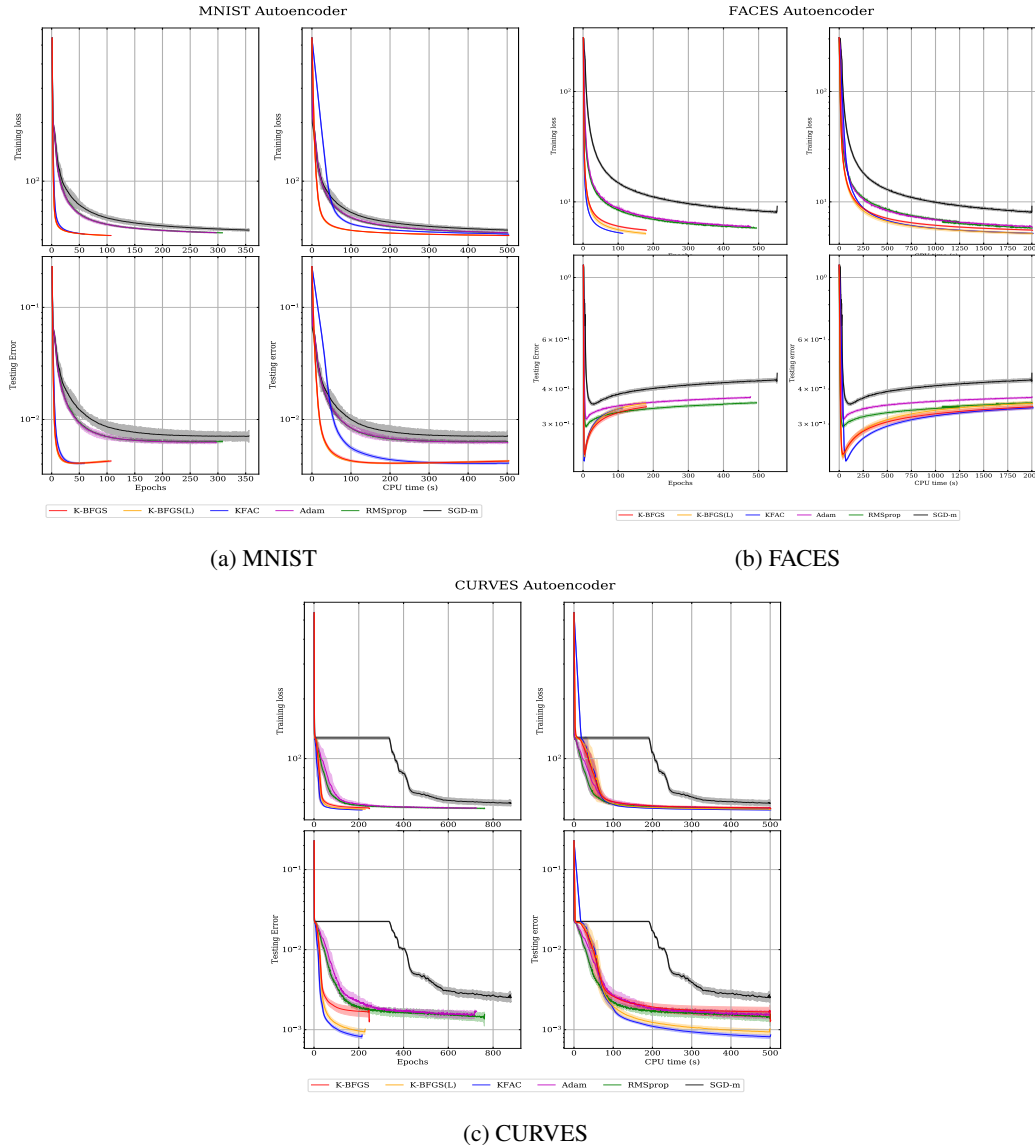

(a) MNIST

(b) FACES

(c) CURVES

Figure 1: Comparison between algorithms on (a) MNIST, (b) FACES, (c) CURVES. For each problem, the upper (lower) rower depicts training loss (testing (mean square) error), whereas the left (right) column depicts training/test progress versus epoch (CPU time), respectively. After each epoch, the training loss/testing error from the whole training/testing set is reported (the time for computing this loss is not included in the plots). For each problem, algorithms are terminated after the same amount of CPU time.

To obtain the results in Figure 1, we first did a grid-search on (learning rate, damping) pairs for all algorithms (except for SGD-m, whose grid-search was only on learning rate), where damping refers to $\lambda$ for K-BFGS/K-BFGS(L)/KFAC, and $\epsilon$ for RMSprop/Adam. We then selected the best (learning rate, damping) pairs with the lowest training loss encountered. The range for the grid-search and the best HP values (as well as other fixed HP values) are listed in Section D in the appendix. Using the

best HP values that we found, we then made 20 runs employing different random seeds, and plotted the mean value of the 20 runs as the solid line and the standard deviation as the shaded area.[1]

From the training loss plots in Figure 1, our algorithms clearly outperformed the first-order methods, except for RMSprop/Adam on CURVES, with respect to CPU time, and performed comparably to KFAC in terms of both CPU time and number of epochs taken. The testing error plots in Figure 1 show that our K-BFGS(L) method and KFAC behave very similarly and substantially outperform all of the first-order methods in terms of both of these measures. This suggests that our algorithms not only optimize well, but also generalize well.

To further demonstrate the robustness of our algorithms, we examined the loss under various HP settings, which showed that our algorithms are stable under a fairly wide range for the HPs (see Section D.4 in the appendix).

We also repeated our experiments using mini-batches of size 100 for all algorithms (see Figures 4, 5, and 6 in the appendix, where HPs are optimally tuned for batch sizes of 100) and our proposed methods continue to demonstrate advantageous performance, both in training and testing. For these experiments, we did not experiment with 20 random seeds. These results show that our approach works as well for relatively small mini-batch sizes of 100, as those of size 1000, which are less noisy, and hence is robust in the stochastic setting employed to train DNNs.

Compared with other methods mentioned in this paper, our K-BFGS and K-BFGS(L) have the extra advantage of being able to double the size of minibatch for computing the stochastic gradient with almost no extra cost, which might be of particular interest in a highly stochastic setting. See Section D.6 in the appendix for more discussion on this and some preliminary experimental results.

## 7 Conclusion

We proposed Kronecker-factored QN methods, namely, K-BFGS and K-BFGS(L), for training multi-layer feed-forward neural network models, that use layer-wise second-order information and require modest memory and computational resources. Experimental results indicate that our methods outperform or perform comparably to the state-of-the-art first-order and second-order methods. Our methods can also be extended to convolutional and recurrent NNs.

## Broader Impact

The research presented in this paper provides a new method for training DNNs that our experimental testing has shown to be more efficient in several cases than current state-of-the-art optimization methods for this task. Consequently, because of the wide use of DNNs in machine learning, this should help save a substantial amount of energy. Our new algorithms simply attempt to minimize the loss function that are given to it and the computations that it performs are all transparent. In machine learning, DNNs can be trained to address many types of problems, some of which should lead to positive societal outcomes, such as ones in medicine (e.g., diagnostics and drug effectiveness), autonomous vehicle development, voice recognition and climate change. Of course, optimization algorithms for training DNNs can be used to train models that may have negative consequences, such as those intended to develop psychological profiles, invade privacy and justify biases. The misuse of any efficient optimization algorithm for machine learning, and in particular our algorithms, is beyond the control of the work presented here.

## Acknowledgments and Disclosure of Funding

DG and YR were supported in part by NSF Grant IIS-1838061. DG acknowledges the computational support provided by Google Cloud Platform Education Grant, Q81G-U4X3-5AG5-F5CG.

## Footnotes

[1]Results were obtained on a machine with 8 x Intel(R) Xeon(R) CPU @ 2.30GHz and 1 x NVIDIA Tesla P100. Code is available at `https://github.com/renyiryry/kbfgs_neurips2020_public`.

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
