[Supplementary Material]

# A  Pseudocode for K-BFGS/K-BFGS(L)

Algorithm 4 gives pseudocode for K-BFGS/K-BFGS(L), which is implemented in the experiments. For details see Sections 3, 4, and Section C in the Appendix.

---

**Algorithm 4** Pseudocode for K-BFGS / K-BFGS(L)

---

**Require:** Given initial weights $\theta = \left[ \text{vec} \left( W_1 \right)^\top, \ldots, \text{vec} \left( W_L \right)^\top \right]^\top$, batch size $m$, learning rate $\alpha$, damping value $\lambda$, and for K-BFGS(L), the number of $(\mathbf{s}, \mathbf{y})$ pairs $p$ that are stored and used to compute $H_g^l$ at each iteration

1: $\mu_1 = 0.2$, $\beta = 0.9$ {set default hyper-parameter values}
2: $\lambda_A = \lambda_G = \sqrt{\lambda}$ {split the damping into $A$ and $G$}
3: $\widehat{\nabla \mathbf{f}}_l = 0$, $A_l = \mathbb{E}_i \left[ \mathbf{a}_{l-1}(i) \mathbf{a}_{l-1}(i)^\top \right]$ by forward pass, $H_a^l = (A_l + \lambda_A I_A)^{-1}$, $H_g^l = I$ $(l = 1, ..., L)$ {Initialization}
4: **for** $k = 1, 2, \ldots$ **do**
5:     Sample mini-batch of size $m$: $M_k = \{\xi_{k,i}, i = 1, \ldots, m\}$
6:     Perform a forward-backward pass over the current mini-batch $M_k$ to compute $\overline{\nabla \mathbf{f}}_l$, $\mathbf{a}_l$, $\mathbf{h}_l$, and $\mathbf{g}_l$ $(l = 1, \ldots, L)$ (see Algorithm 1)
7:     **for** $l = 1, \ldots, L$ **do**
8:         $\widehat{\nabla \mathbf{f}}_l = \beta \widehat{\nabla \mathbf{f}}_l + (1 - \beta) \overline{\nabla \mathbf{f}}_l$
9:         $p_l = H_g^l \widehat{\nabla \mathbf{f}}_l H_a^l$
10:         {In K-BFGS(L), when computing $H_g^l \left( \widehat{\nabla \mathbf{f}}_l H_a^l \right)$, L-BFGS is initialized with an identity matrix}
11:         $W_l = W_l - \alpha \cdot p_l$
12:     Perform another forward-backward pass over $M_k$ to compute $\mathbf{h}_l^+$, $\mathbf{g}_l^+$ $(l = 1, \ldots, L)$
13:     **for** $l = 1, ..., L$ **do**
14:         {Use damped BFGS or L-BFGS to update $H_g^l$ (see Section 3)}
15:         $\mathbf{s}_g^l = \beta \cdot \mathbf{s}_g^l + (1 - \beta) \cdot \left( \overline{\mathbf{h}_l^+} - \overline{\mathbf{h}_l} \right)$, $\mathbf{y}_g^l = \beta \cdot \mathbf{y}_g^l + (1 - \beta) \cdot \left( \overline{\mathbf{g}_l^+} - \overline{\mathbf{g}_l} \right)$
16:         $(\tilde{\mathbf{s}}_g^l, \tilde{\mathbf{y}}_g^l) = \text{DD}(\mathbf{s}_g^l, \mathbf{y}_g^l)$ with $H = H_g^l$, $\mu_1 = \mu_1$, $\mu_2 = \lambda_G$ {See Algorithm 3}
17:         Use BFGS or L-BFGS with $(\tilde{\mathbf{s}}_g^l, \tilde{\mathbf{y}}_g^l)$ to update $H_g^l$
18:         {Use Hessian-action BFGS to update $H_a^l$ (see Section 4)}
19:         $A_l = \beta \cdot A_l + (1 - \beta) \cdot \overline{\mathbf{a}_{l-1} \mathbf{a}_{l-1}^\top}$
20:         $A_l^{\text{LM}} = A_l + \lambda_A I_A$
21:         $\mathbf{s}_a^l = H_a^l \cdot \overline{\mathbf{a}_{l-1}}$, $\mathbf{y}_a^l = A_l^{\text{LM}} \mathbf{s}_a^l$
22:         Use BFGS with $(\mathbf{s}_a^l, \mathbf{y}_a^l)$ to update $H_a^l$

---

# B  Convergence: Proofs of Lemmas 1-3 and Theorem 2

In this section, we prove the convergence of Algorithm 5, a variant of K-BFGS(L). Algorithm 5 is very similar to our actual implementation of K-BFGS(L) (i.e. Algorithm 4), except that

- we skip updating $H_g^l$ if $(\tilde{\mathbf{s}}_g^l)^\top \tilde{\mathbf{y}}_g^l < \mu_1 (\tilde{\mathbf{y}}_g^l)^\top H_g^l \tilde{\mathbf{y}}_g^l$ (see Line 16);
- we set $H_a^l$ to the exact inverse of $A_l^{\text{LM}}$ (see Line 21);
- we use decreasing step sizes $\{\alpha_k\}$ as specified in Theorem 2;
- we use the mini-batch gradient instead of the momentum gradient (see Line 8).

To accomplish this, we prove Lemmas 1-3, which in addition to Assumptions AS.1-2, ensure that all of the assumptions in Theorem 2.8 in [41] are satisfied, and hence that the generic stochastic quasi-Newton (SQN) method, i.e. Algorithm 6, below converges. Specifically, Theorem 2.8 in [41] requires, in addition to Assumptions AS.1-2, the assumption

**AS. 4.** *There exist two positive constants $\underline{\kappa}, \bar{\kappa}$, such that $\underline{\kappa} I \preceq H_k \preceq \bar{\kappa} I, \forall k$; for any $k \geq 2$, the random variable $H_k$ depends only on $\xi_{[k-1]}$.*

**Algorithm 5** K-BFGS(L) with DD-skip and exact inversion of $A_l^{\text{LM}}$

---

**Require:** Given initial weights $\theta = \left[\text{vec}\left(W_1\right)^{\top}, \ldots, \text{vec}\left(W_L\right)^{\top}\right]^{\top}$, batch size $m$, learning rate $\{\alpha_k\}$, damping value $\lambda$, and the number of $(\mathbf{s}, \mathbf{y})$ pairs $p$ that are stored and used to compute $H_g^l$ at each iteration

1: $\mu_1 = 0.2$, $\beta = 0.9$ {set default hyper-parameter values}
2: $\lambda_A = \lambda_G = \sqrt{\lambda}$ {split the damping into $A$ and $G$}
3: $A_l(0) = \mathbb{E}_i\left[\mathbf{a}_{l-1}(i)\mathbf{a}_{l-1}(i)^{\top}\right]$ by forward pass, $H_a^l(0) = (A_l(0) + \lambda_A I_A)^{-1}$, $H_g^l(0) = I$ $(l = 1, ..., L)$ {Initialization}
4: **for** $k = 1, 2, \ldots$ **do**
5:     Sample mini-batch of size $m$: $M_k = \{\xi_{k,i}, i = 1, \ldots, m\}$
6:     Perform a forward-backward pass over the current mini-batch $M_k$ to compute $\overline{\nabla}\mathbf{f}_l$, $\mathbf{a}_l$, $\mathbf{h}_l$, and $\mathbf{g}_l$ $(l = 1, \ldots, L)$ (see Algorithm 1)
7:     **for** $l = 1, \ldots, L$ **do**
8:         $p_l = H_g^l(k-1)\widehat{\nabla}\mathbf{f}_l H_a^l(k-1)$, where $\widehat{\nabla}\mathbf{f}_l = \overline{\nabla}\mathbf{f}_l$
9:         {When computing $H_g^l\left(\widehat{\nabla}\mathbf{f}_l H_a^l\right)$, L-BFGS is initialized with an identity matrix}
10:         $W_l = W_l - \alpha_k \cdot p_l$
11:     Perform another forward-backward pass over $M_k$ to compute $\mathbf{h}_l^+$, $\mathbf{g}_l^+$ $(l = 1, \ldots, L)$
12:     **for** $l = 1, ..., L$ **do**
13:         {Use damped L-BFGS with skip to update $H_g^l$ (see Section 3)}
14:         $\mathbf{s}_g^l = \beta \cdot \mathbf{s}_g^l + (1 - \beta) \cdot \left(\overline{\mathbf{h}_l^+} - \overline{\mathbf{h}_l}\right)$, $\mathbf{y}_g^l = \beta \cdot \mathbf{y}_g^l + (1 - \beta) \cdot \left(\overline{\mathbf{g}_l^+} - \overline{\mathbf{g}_l}\right)$
15:         $(\tilde{\mathbf{s}}_g^l, \tilde{\mathbf{y}}_g^l) = \text{DD}(\mathbf{s}_g^l, \mathbf{y}_g^l)$ with $H = H_g^l(k-1)$, $\mu_1 = \mu_1$, $\mu_2 = \lambda_G$ {See Algorithm 3}
16:         **if** $(\tilde{\mathbf{s}}_g^l)^{\top}\tilde{\mathbf{y}}_g^l \geq \mu_1 (\tilde{\mathbf{y}}_g^l)^{\top} H_g^l \tilde{\mathbf{y}}_g^l$ **then**
17:             Use L-BFGS with $(\tilde{\mathbf{s}}_g^l, \tilde{\mathbf{y}}_g^l)$ to update $H_g^l(k)$
18:         {Use exact inversion to compute $H_a^l$}
19:         $A_l(k) = \beta \cdot A_l(k-1) + (1-\beta) \cdot \mathbf{a}_{l-1}\mathbf{a}_{l-1}^{\top}$
20:         $A_l^{\text{LM}}(k) = A_l(k) + \lambda_A I_A$
21:         $H_a^l(k) = \left(A_l^{\text{LM}}(k)\right)^{-1}$

---

**Algorithm 6** SQN method for nonconvex stochastic optimization.

---

**Require:** Given $\theta_1 \in \mathbb{R}^n$, batch sizes $\{m_k\}_{k \geq 1}$, and step sizes $\{\alpha_k\}_{k \geq 1}$

1: **for** $k = 1, 2, \ldots$ **do**
2:     Calculate $\widehat{\nabla}\mathbf{f}_k = \frac{1}{m_k}\sum_{i=1}^{m_k} \nabla\mathbf{f}(\theta_k, \xi_{k,i})$
3:     Generate a positive definite Hessian inverse approximation $H_k$
4:     Calculate $\theta_{k+1} = \theta_k - \alpha_k H_k \widehat{\nabla}\mathbf{f}_k$

---

In the following proofs, $\|\cdot\|$ denotes the 2-norm for vectors, and the spectral norm for matrices.

**Proof of Lemma 1:**

*Proof.* Because $A_l^{\text{LM}}(k) \succeq \lambda_A I_A$, we have that $H_a^l(k) \preceq \bar{\kappa}_a I_A$, where $\bar{\kappa}_a = \frac{1}{\lambda_A}$.

On the other hand, for any $\mathbf{x} \in \mathbb{R}^{d_l}$, by Cauchy-Schwarz, $\langle \mathbf{a}_{l-1}(i), \mathbf{x}\rangle^2 \leq \|\mathbf{x}\|^2\|\mathbf{a}_{l-1}(i)\|^2 \leq \|\mathbf{x}\|^2(1+\varphi^2 d_l)$. Hence, $\left\|\mathbf{a}_{l-1}\mathbf{a}_{l-1}^{\top}\right\| \leq 1 + \varphi^2 d_l$; similarly, $\|A_l(0)\| \leq 1 + \varphi^2 d_l$. Because $\|A_l(k)\| \leq \beta\|A_l(k-1)\| + (1 - \beta)\left\|\overline{\mathbf{a}_{l-1}\mathbf{a}_{l-1}^{\top}}\right\|$, by induction, $\|A_l(k)\| \leq 1 + \varphi^2 d_l$ for any $k$ and $l$. Thus, $\|A_l^{\text{LM}}(k)\| \leq 1 + \varphi^2 d_l + \lambda_A$. Hence, $H_a^l(k) \succeq \underline{\kappa}_a I_A$, where $\underline{\kappa}_a = \frac{1}{1+\varphi^2 d_l + \lambda_A}$.

$\square$

**Proof of Lemma 2:**

*Proof.* To simplify notation, we omit the subscript $g$, superscript $l$ and the iteration index $k$ in the proof. Hence, our goal is to prove $\underline{\kappa}_g I \preceq H = H_g^l(k) \preceq \bar{\kappa}_g I$, for any $l$ and $k$. Let $(\mathbf{s}_i, \mathbf{y}_i)$ $(i = 1, ..., p)$ denote the pairs used in an L-BFGS computation of $H$. Since $(\mathbf{s}_i, \mathbf{y}_i)$ was **not skipped**, $\frac{\mathbf{y}_i^\top \bar{H}^{(i)} \mathbf{y}_i}{\mathbf{s}_i^\top \mathbf{y}_i} \leq \frac{1}{\mu_1}$, where $\bar{H}^{(i)}$ denotes the matrix $H_g^l$ used at the iteration in which $\mathbf{s}_i$ and $\mathbf{y}_i$ were computed. Note that this is not the matrix $H_i$ used in the recursive computation of $H$ at the current iterate $\theta_k$.

Given an initial estimate $H_0 = B_0^{-1} = I$ of $(G_g^l(\theta_k))^{-1}$, the L-BFGS method updates $H_i$ recursively as

$$H_i = \left(I - \rho_i \mathbf{s}_i \mathbf{y}_i^\top\right) H_{i-1} \left(I - \rho_i \mathbf{y}_i \mathbf{s}_i^\top\right) + \rho_i \mathbf{s}_i \mathbf{s}_i^\top, \quad i = 1, \ldots, p, \tag{12}$$

where $\rho_i = (\mathbf{s}_i^\top \mathbf{y}_i)^{-1}$, and equivalently,

$$B_i = B_{i-1} + \frac{\mathbf{y}_i \mathbf{y}_i^\top}{\mathbf{s}_i^\top \mathbf{y}_i} - \frac{B_{i-1} \mathbf{s}_i \mathbf{s}_i^\top B_{i-1}}{\mathbf{s}_i^\top B_{i-1} \mathbf{s}_i}, \quad i = 1, \ldots, p,$$

where $B_i = H_i^{-1}$. Since we use DD with skipping, we have that $\frac{\mathbf{s}_i^\top \mathbf{s}_i}{\mathbf{s}_i^\top \mathbf{y}_i} \leq \frac{1}{\mu_2}$ and $\frac{\mathbf{y}_i^\top \bar{H}^{(i)} \mathbf{y}_i}{\mathbf{s}_i^\top \mathbf{y}_i} \leq \frac{1}{\mu_1}$. Note that we don't have $\frac{\mathbf{y}_i^\top H_{i-1} \mathbf{y}_i}{\mathbf{s}_i^\top \mathbf{y}_i} \leq \frac{1}{\mu_1}$, so we cannot direct apply (10). Hence, by (8), we have that $||B_i|| \leq ||B_{i-1}|| \left(1 + \frac{1}{\mu_1}\right)$. Hence, $||B|| = ||B_p|| \leq ||B_0|| \left(1 + \frac{1}{\mu_1}\right)^p = \left(1 + \frac{1}{\mu_1}\right)^p$. Thus, $B \preceq \left(1 + \frac{1}{\mu_1}\right)^p I$, $H \succeq \left(1 + \frac{1}{\mu_1}\right)^{-p} I := \underline{\kappa}_g I$.

On the other hand, since $\underline{\kappa}_g$ is a uniform lower bound for $H_g^l(k)$ for any $k$ and $l$, $\bar{H}^{(i)} \succeq \underline{\kappa}_g I$. Thus,

$$\frac{1}{\mu_1} \geq \frac{\mathbf{y}_i^\top \bar{H}^{(i)} \mathbf{y}_i}{\mathbf{s}_i^\top \mathbf{y}_i} \geq \underline{\kappa}_g \frac{\mathbf{y}_i^\top \mathbf{y}_i}{\mathbf{s}_i^\top \mathbf{y}_i} \Rightarrow \frac{\mathbf{y}_i^\top \mathbf{y}_i}{\mathbf{s}_i^\top \mathbf{y}_i} \leq \frac{1}{\mu_1 \underline{\kappa}_g}.$$

Hence, using the fact that $||uv^\top|| = ||u|| \cdot ||v||$ for any vectors $u, v$, $||\rho_i \mathbf{s}_i \mathbf{s}_i^\top|| = \rho_i ||\mathbf{s}_i|| ||\mathbf{s}_i|| = \frac{\mathbf{s}_i^\top \mathbf{s}_i}{\mathbf{s}_i^\top \mathbf{y}_i} \leq \frac{1}{\mu_2}$,

$$
\begin{aligned}
||H_i|| &= || \left(I - \rho_i \mathbf{s}_i \mathbf{y}_i^\top\right) H_{i-1} \left(I - \rho_i \mathbf{y}_i \mathbf{s}_i^\top\right) + \rho_i \mathbf{s}_i \mathbf{s}_i^\top || \\
&= ||H_{i-1} + \rho_i^2 (\mathbf{y}_i^\top H_{i-1} \mathbf{y}_i) \mathbf{s}_i \mathbf{s}_i^\top - \rho_i \mathbf{s}_i \mathbf{y}_i^\top H_{i-1} - \rho_i H_{i-1} \mathbf{y}_i \mathbf{s}_i^\top + \rho_i \mathbf{s}_i \mathbf{s}_i^\top || \\
&\leq ||H_{i-1}|| + ||\rho_i^2 (\mathbf{y}_i^\top H_{i-1} \mathbf{y}_i) \mathbf{s}_i \mathbf{s}_i^\top || + ||\rho_i \mathbf{s}_i \mathbf{y}_i^\top H_{i-1}|| + ||\rho_i H_{i-1} \mathbf{y}_i \mathbf{s}_i^\top|| + ||\rho_i \mathbf{s}_i \mathbf{s}_i^\top || \\
&\leq ||H_{i-1}|| + ||H_{i-1}|| \cdot ||\rho_i^2 (\mathbf{y}_i^\top \mathbf{y}_i) \mathbf{s}_i \mathbf{s}_i^\top || + 2\rho_i ||\mathbf{s}_i|| \cdot ||\mathbf{y}_i^\top H_{i-1}|| + \frac{1}{\mu_2} \\
&\leq ||H_{i-1}|| + ||H_{i-1}|| \cdot \frac{1}{\mu_1 \underline{\kappa}_g} \frac{1}{\mu_2} + 2\rho_i ||\mathbf{s}_i|| \cdot ||\mathbf{y}_i^\top|| \cdot ||H_{i-1}|| + \frac{1}{\mu_2} \\
&\leq ||H_{i-1}|| \left(1 + \frac{1}{\mu_1 \underline{\kappa}_g} \frac{1}{\mu_2} + 2\frac{1}{\sqrt{\mu_1 \mu_2 \underline{\kappa}_g}}\right) + \frac{1}{\mu_2} \\
&= \hat{\mu} ||H_{i-1}|| + \frac{1}{\mu_2}, \quad \text{where} \quad \hat{\mu} = \left(1 + \frac{1}{\sqrt{\mu_1 \mu_2 \underline{\kappa}_g}}\right)^2.
\end{aligned}
$$

From the fact that $H_0 = I$, and induction, we have that $||H|| \leq \hat{\mu}^p + \frac{\hat{\mu}^p - 1}{\hat{\mu} - 1} \frac{1}{\mu_2} \equiv \bar{\kappa}_g$.

$\square$

**Proof of Lemma 3:**

*Proof.* By Lemma 1, 2 and the fact that $H_k = \text{diag}\{H_a^1(k-1) \otimes H_g^1(k-1), ..., H_a^L(k-1) \otimes H_g^L(k-1)\}$. $\square$

**Proof of Theorem 2:**

*Proof.* To show that Algorithm 5 lies in the framework of Algorithm 6, it suffices to show that $H_k$ generated by Algorithm 5 is positive definite, which is true since $H_k = \text{diag}\{H_a^1(k-1) \otimes H_g^1(k-1), ..., H_a^L(k-1) \otimes H_g^L(k-1)\}$ and $H_a^l(k)$ and $H_g^l(k)$ are positive definite for all $k$ and $l$. Then by Lemma 3, and the fact that $H_k$ depends on $H_a^l(k-1)$ and $H_g^l(k-1)$, and $H_a^l(k-1)$ and $H_g^l(k-1)$ does not depend on random samplings in the $k$th iteration, AS.4 holds. Hence, Theorem 2.8 of [41] applies to Algorithm 5, proving Theorem 2. $\square$

## C  Powell's Damped BFGS Updating

For BFGS and L-BFGS, one needs $\mathbf{y}^\top \mathbf{s} > 0$. However, when used to update $H_g^l$, there is no guarantee that $(\mathbf{y}_g^l)^\top \mathbf{s}_g^l > 0$ for any layer $l = 1, \ldots, L$. In deterministic optimization, positive definiteness of the QN Hessian approximation $B$ (or its inverse) is maintained by performing an inexact line search that ensures that $\mathbf{s}^T \mathbf{y} > 0$, which is always possible as long as the function being minimized is bounded below. However, this would be expensive to do for DNN. Thus, we propose the following heuristic based on Powell's damped-BFGS approach [35].

**Powell's Damping on $B$.**  Powell's damping on $B$, proposed in [35], replaces $\mathbf{y}$ in the BFGS update, by $\tilde{\mathbf{y}} = \theta \mathbf{y} + (1 - \theta) B \mathbf{s}$, where

$$\theta = \begin{cases} \frac{(1-\mu)\mathbf{s}^\top B \mathbf{s}}{\mathbf{s}^\top B \mathbf{s} - \mathbf{s}^\top \mathbf{y}}, & \text{if } \mathbf{s}^\top \mathbf{y} < \mu \mathbf{s}^\top B \mathbf{s}, \\ 1, & \text{otherwise.} \end{cases}$$

It is easy to verify that $\mathbf{s}^\top \tilde{\mathbf{y}} \geq \mu \mathbf{s}^\top B \mathbf{s}$.

**Powell's Damping on $H$.**  In Powell's damping on $H$ (see e.g. [3]), $\tilde{\mathbf{s}} = \theta \mathbf{s} + (1 - \theta) H \mathbf{y}$ replaces $\mathbf{s}$, where

$$\theta = \begin{cases} \frac{(1-\mu_1)\mathbf{y}^\top H \mathbf{y}}{\mathbf{y}^\top H \mathbf{y} - \mathbf{s}^\top \mathbf{y}}, & \text{if } \mathbf{s}^\top \mathbf{y} < \mu_1 \mathbf{y}^\top H \mathbf{y}, \\ 1, & \text{otherwise.} \end{cases}$$

This is used in lines 2 and 3 of the DD (Algorithm 3). It is also easy to verify that $\tilde{\mathbf{s}}^\top \mathbf{y} \geq \mu_1 \mathbf{y}^\top H \mathbf{y}$.

**Powell's Damping with $B = I$.**  Powell's damping on $B$ is not suitable for our algorithms because we do not keep track of $B$. Moreover, it does not provide a simple bound on $\frac{\tilde{\mathbf{s}}^\top \tilde{\mathbf{s}}}{\tilde{\mathbf{s}}^\top \tilde{\mathbf{y}}}$ that is independent of $\|B\|$. Therefore, we use Powell's damping with $B = I$, in lines 4 and 5 of the DD (Algorithm 3). It is easy to verify that it ensures that $\tilde{\mathbf{s}}^\top \tilde{\mathbf{y}} \geq \mu_2 \tilde{\mathbf{s}}^\top \tilde{\mathbf{s}}$.

Powell's damping with $B = I$ can be interpreted as adding an Levenberg-Marquardt (LM) damping term to $B$. Note that an LM damping term $\mu_2$ would lead to $B \succeq \mu_2 I$. Then, the secant condition $\tilde{\mathbf{y}} = B\tilde{\mathbf{s}}$ implies

$$\tilde{\mathbf{y}}^\top \tilde{\mathbf{s}} = \tilde{\mathbf{s}}^\top B \tilde{\mathbf{s}} \geq \mu_2 \tilde{\mathbf{s}}^\top \tilde{\mathbf{s}},$$

which is the same inequality as we get using Powell's damping with $B = I$. Note that although the $\mu_2$ parameter in Powell's damping with $B = I$ can be interpreted as an LM damping, we recommend setting the value of $\mu_2$ within $(0, 1]$ so that $\tilde{\mathbf{y}}$ is a convex combination of $\mathbf{y}$ and $\tilde{\mathbf{s}}$. In all of our experimental tests, we found that the best value for the hyperparameter $\lambda$ for both K-BFGS and K-BFGS(L) was less than or equal to 1, and hence that $\mu_2 = \lambda_G = \sqrt{\lambda}$ was in the interval $(0, 1]$.

### C.1  Double Damping (DD)

Our double damping (Algorithm 3) is a two-step damping procedure, where the first step (i.e. Powell's damping on $H$) can be viewed as an interpolation between the current curvature and the previous ones, and the second step (i.e. Powell's damping with $B = I$) can be viewed as an LM damping.

Recall that there is no guarantee that $\frac{\mathbf{y}^\top H \mathbf{y}}{\mathbf{s}^\top \mathbf{y}} \leq \frac{2}{\mu_1}$ holds after DD. While we skip using pairs that do not satisfy this inequality, when updating $H_g^l$ in proving the convergence of the K-BFGS(L) variant Algorithm 5 , we use all $(\mathbf{s}, \mathbf{y})$ pairs to update $H_g^l$ in our implementations of both K-BFGS and K-BFGS(L) . However, whether one skips or not makes only slight difference in the performance of these algorithms, because as our empirical testing has shown, at least 90% of the iterations satisfy $\frac{\mathbf{y}^\top H \mathbf{y}}{\mathbf{s}^\top \mathbf{y}} \leq \frac{2}{\mu_1}$, even if we don't skip. See Figure 2 which reports results on this for K-BFGS(L) when tested on the MNIST, FACES and CURVES datasets.

## D  Implementation Details and More Experiments

### D.1  Description of Competing Algorithms

#### D.1.1  KFAC

We first describe KFAC in Algorithm 7. Note that $G_l$ in KFAC refers to the $G$ matrices in [30], which is different from the $G_l$ in K-BFGS.

Figure 2: Fraction of the number of iterations in each epoch, in which the inequality $\frac{\mathbf{y}^\top H \mathbf{y}}{\mathbf{s}^\top \mathbf{y}} \leq \frac{2}{\mu_1}$ holds (upper plots), and the average value of $\frac{\mathbf{y}^T H \mathbf{y}}{\mathbf{s}^\top \mathbf{y}}$ (lower plots) in each epoch. Legends in each plot assign different colors to represent each layer $l$.

### D.1.2 Adam/RMSprop

We implement Adam and RMSprop exactly as in [24] and [21], respectively. Note that the only difference between them is that Adam does bias correction for the 1st and 2nd moments of gradient while RMSprop does not.

### D.1.3 Initialization of Algorithms

We now describe how each algorithm is initialized. For all algorithms, $\widehat{\nabla \mathbf{f}}$ is always initialized as zero.

For second-order information, we use a "warm start" to estimate the curvature when applicable. In particular, we estimate the following curvature information using the entire training set before we start updating parameters. The information gathered is

- $A_l$ for K-BFGS and K-BFGS(L);
- $A_l$ and $G_l$ for KFAC;
- $\nabla f \odot \nabla f$ for RMSprop;

**Algorithm 7 KFAC**

---

**Require:** Given $\theta_0$, batch size $m$, and learning rate $\alpha$, damping value $\lambda$, inversion frequency $T$
 1: **for** $k = 1, 2, \ldots$ **do**
 2:    Sample mini-batch of size $m$: $M_k = \{\xi_{k,i}, i = 1, \ldots, m\}$
 3:    Perform a forward-backward pass over the current mini-batch $M_k$ (see Algorithm 1)
 4:    **for** $l = 1, 2, \ldots L$ **do**
 5:       $p_l = H_g^l \widehat{\nabla \mathbf{f}}_l H_a^l$
 6:       $W_l = W_l - \alpha \cdot p_l.$
 7:    Perform another pass over $M_k$ with $y$ sampled from the predictive distribution
 8:    Update $A_l = \beta \cdot A_l + (1 - \beta) \cdot \overline{\mathbf{a}_{l-1}\mathbf{a}_{l-1}^\top}$, $G_l = \beta \cdot G_l + (1 - \beta) \cdot \overline{\mathbf{g}_l \mathbf{g}_l^\top}$
 9:    **if** $k \leq T$ or $k \equiv 0 \pmod{T}$ **then**
10:       Recompute $H_a^l = (A_l + \sqrt{\lambda}I)^{-1}$, $H_g^l = (G_l + \sqrt{\lambda}I)^{-1}$

---

- Not applicable to Adam because of the bias correction.

Lastly, for K-BFGS and K-BFGS(L), $H_a^l$ is always initially set to an identity matrix. $H_g^l$ is also initially set to an identity matrix in K-BFGS; for K-BFGS(L), when updating $H_g^l$ using L-BFGS, the incorporation of the information from the $p$ $(\mathbf{s}, \mathbf{y})$ pairs is applied to an initial matrix that is set to an identity matrix. Hence, the above initialization/warm start costs are roughly twice as large for KFAC as they are for K-BFGS and K-BFGS(L).

## D.2   Autoencoder Problems

Table 3 lists information about the three datasets, namely, MNIST[2], FACES[3], and CURVES[4]. Table 4 specifies the architecture of the 3 problems, where binary entropy $\mathcal{L}(a_L, y) = \sum_n [y_n \log a_{L,n} + (1 - y_n) \log(1 - a_{L,n})]$, MSE $\mathcal{L}(a_L, y) = \frac{1}{2}\sum_n (a_{L,n} - y_n)^2$. Besides the loss function in Table 4, we further add a regularization term $\frac{\eta}{2}||\theta||^2$ to the loss function, where $\eta = 10^{-5}$.

Table 3: Info for 3 datasets

| Dataset | # data points | # training examples | # testing examples |
|---------|---------------|---------------------|--------------------|
| MNIST   | 70,000        | 60,000              | 10,000             |
| FACES   | 165,600       | 103,500             | 62,100             |
| CURVES  | 30,000        | 20,000              | 10,000             |

Table 4: Architecture of 3 auto-encoder problems

| Dataset | Layer width & activation | Loss function |
|---------|--------------------------|---------------|
| MNIST | [784, 1000, 500, 250, 30, 250, 500, 1000, 784]<br>[ReLU, ReLU, ReLU, linear, ReLU, ReLU, ReLU, sigmoid] | binary entropy |
| FACES | [625, 2000, 1000, 500, 30, 500, 1000, 2000, 625]<br>[ReLU, ReLU, ReLU, linear, ReLU, ReLU, ReLU, linear] | MSE |
| CURVES | [784, 400, 200, 100, 50, 25, 6, 25, 50, 100, 200, 400, 784]<br>[ReLU, ReLU, ReLU, ReLU, ReLU, linear,<br>ReLU, ReLU, ReLU, ReLU, ReLU, sigmoid] | binary entropy |

## D.3   Specification of Hyper-parameters

In our experiments, we focus our tuning effort onto learning rate and damping. The range of the tuning values is listed below:

- learning rate $\alpha_k = \alpha \in$ { 1e-5, 3e-5, 1e-4, 3e-4, 1e-3, 3e-3, 1e-2, 3e-2, 1e-1, 3e-1, 1, 3, 10 }.

- damping:

  - $\lambda$ for K-BFGS, K-BFGS(L) and KFAC: $\lambda \in$ { 3e-3, 1e-2, 3e-2, 1e-1, 3e-1, 1, 3 }.

  - $\epsilon$ for RMSprop and Adam: $\epsilon \in$ { 1e-10, 1e-8, 1e-6, 1e-4, 1e-3, 1e-2, 1e-1 }.

  - Not applicable for SGD with momentum.

Table 5: Best (learning rate, damping) for Figure 1

|  | K-BFGS | K-BFGS(L) | KFAC | Adam | RMSprop | SGD-m |
|---|---|---|---|---|---|---|
| MNIST | (0.3, 0.3) | (0.3, 0.3) | (1, 3) | (1e-4, 1e-4) | (1e-4, 1e-4) | (0.03, -) |
| FACES | (0.1, 0.1) | (0.1, 0.1) | (0.1, 0.1) | (1e-4, 1e-4) | (1e-4, 1e-4) | (0.01, -) |
| CURVES | (0.1, 0.01) | (0.3, 0.3) | (0.3, 0.3) | (1e-3, 1e-3) | (1e-3, 1e-3) | (0.1, -) |

The best hyper-parameters were those that produced the lowest value of the deterministic loss function encountered at the end of every epoch until the algorithm was terminated. These values were used in Figure 1 and are listed in Table 5. Besides the tuning hyper-parameters, we also list other fixed hyper-parameters with their values:

- Size of minibatch $m = 1000$, which is also suggested in [5].

- Decay parameter:

  - K-BFGS, K-BFGS(L): $\beta = 0.9$;

  - KFAC: $\beta = 0.9$;

  - RMSprop, Adam: Following the notation in [24], we use $\beta_1 = \beta_2 = 0.9$;[5]

  - SGD with momentum: $\beta = 0.9$.

- Other:

  - $\mu_1 = 0.2$ in double damping (DD):

    We recommend to leave the value as default because $\mu_1$ represents the "ratio" between current and past, which is scaling invariant;

  - Number of $(\mathbf{s}, \mathbf{y})$ pairs stored for K-BFGS(L) $p = 100$:

    It might be more efficient to use a smaller $p$ for the narrow layers. We didn't investigate this for simplicity and consistency;

  - Inverse frequency $T = 20$ in KFAC.

## D.4 Sensitivity to Hyper-parameters

Figure 3: Landscape of loss w.r.t hyper-parameters (i.e. learning rate and damping). The left, middle, right columns depict results for MNIST, FACES, CURVES, which are terminated after 500, 2000, 500 seconds (CPU time), respectively, for K-BFGS (upper) and K-BFGS(L) (lower) row.

Figure 3 shows the sensitivity of K-BFGS and K-BFGS(L) to hyper-parameter values (i.e. learning rate and damping). The $x$-axis corresponds to the learning rate $\alpha$, while the $y$-axis correspond to the damping value $\lambda$. Color corresponds to the loss after a certain amount of CPU time. We can see that both K-BFGS and K-BFGS(L) are robust within a fairly wide range of hyper-parameters.

To get the plot, we first obtained training loss with $\alpha \in \{$1e-4, 3e-4, 1e-3, 3e-3, 1e-2, 3e-2, 1e-1, 3e-1, 1$\}$ and $\lambda \in \{$1e-2, 3e-2, 1e-1, 3e-1, 1$\}$, and then drew contour lines of the loss within the above ranges.

## D.5 Experimental Results Using Mini-batches of Size 100

We repeated our experiments using mini-batches of size 100 for all algorithms (see Figures 4, 5, and 6). For each figure, the upper (lower) rower depict training loss (testing (mean square) error), whereas the left (right) column depicts training/test progress versus epoch (CPU time), respectively.

The best hyper-parameters were those that produce the lowest value of the deterministic loss function encountered at the end of every epoch until the algorithm was terminated. These values were used in Figures 4, 5, 6 and are listed in Table 6.

Our proposed methods continue to demonstrate advantageous performance, both in training and testing. It is interesting to note that, whereas for a minibatch size of 1000, KFAC slightly outperformed K-BFGS(L), for a minibatch size of 100, K-BFGS(L) clearly outperformed KFAC in training on CURVES.

Table 6: Best (learning rate, damping) for Figures 4, 5, 6

|  | K-BFGS | K-BFGS(L) | KFAC | Adam | RMSprop | SGD-m |
|---|---|---|---|---|---|---|
| MNIST | (0.1, 0.3) | (0.1, 0.3) | (0.1, 0.3) | (1e-4, 1e-4) | (1e-4, 1e-4) | (0.03, -) |
| FACES | (0.03, 0.03) | (0.03, 0.3) | (0.03, 0.3) | (3e-5, 1e-4) | (3e-5, 1e-4) | (0.01, -) |
| CURVES | (0.3, 1) | (0.3, 0.3) | (0.03, 0.1) | (3e-4, 1e-4) | (3e-3, 1e-4) | (0.03, -) |

MNIST, batch_size=100

Figure 4: Comparison between algorithms on MNIST with batch size 100

FACES, batch_size=100

Figure 5: Comparison between algorithms on FACES with batch size 100

Figure 6: Comparison between algorithms on CURVES with batch size 100

## D.6    Doubling the Mini-batch for the Gradient at Almost No Cost

Compared with other methods mentioned in this paper, our K-BFGS and K-BFGS(L) methods have the extra advantage of being able to double the size of the minibatch used to compute the stochastic gradient with almost no extra cost, which might be of particular interest in a highly stochastic setting. To accomplish this, we can make use of the stochastic gradient $\overline{\nabla \mathbf{f}}^{+}$ computed in the **second** pass of the previous iteration that is needed for computing the $(\overline{\mathbf{s}}, \overline{\mathbf{y}})$ pair for applying the BFGS or L-BFGS updates, and average it with the stochastic gradient $\overline{\nabla \mathbf{f}}$ of the current iteration. For example if the size of minibatch is $m = 1000$, the above "double-grad-minibatch" method computes a stochastic gradient from 2000 data points at each iteration, except at the very first iteration.

The results of some preliminary experiments are depicted in Figure 7, where we compare an earlier version of the K-BFGS algorithm (Algorithm 4), which uses a slightly different variant of Hessian-action to update $H_a^l$, using a size of $m = 1000$ for mini-batches, with its "double-grad-minibatch" variants for $m = 500$ and $1000$. Even though "double-grad" does not help much in these experiments, our K-BFGS algorithm performs stably across these different variants. These results indicate that there is a potential for further improvements; e.g., a finer grid search might identify hyper-parameter values that result in better performing algorithms.

Figure 7: Comparison between K-BFGS and its "double-grad" variants

## Footnotes

[2]Downloadable at http://yann.lecun.com/exdb/mnist/

[3]Downloadable at www.cs.toronto.edu/~jmartens/newfaces_rot_single.mat

[4]Downloadable at www.cs.toronto.edu/~jmartens/digs3pts_1.mat

[5]The default value of $\beta_2$ recommended in [24] is 0.999. Hence, we also tested $\beta_2 = 0.999$, and obtained results that were similar to those presented in Figure 1 (i.e., with $\beta_2 = 0.9$). For the sake of fair comparison, we chose to report the results with $\beta_2 = 0.9$.