[Reviews · NeurIPS 2020]

Review 1

Summary and Contributions: The author proposed a quasi-newton optimization algorithm for solving DNN. Second-order algorithms are not popular as today because the computation cost and storage cost of maintaining the hessian matrix can be prohibitive. The author approximate the hessian matrix using a block diagonal matrix which reduces the cost significantly. They further proposed proper damping method to control the upper / lower bound in BFGS.

Strengths: Due to the popularity of DNN models, any minor improvement in the optimization space can have huge impact. Therefore, I'm very happy to see this work moves second-order optimization forward. It is essentially a challenging problem because the activation function and network structure can be complex. Empirical results show the proposed method works. Yet the techniques are very general. It may benefit more networks than auto-encoders.

Weaknesses: I don't see obvious limitations.

Correctness: Yes

Clarity: Yes. Some minor issues: - "LBFGS" and "L-BFGS" should be unified. - line 68: New BGFS and L-BFGS, should be "BFGS" - the gap between line 77 / 78 is too wide. Double check if adding authors will resolve it

Relation to Prior Work: Yes. The proof of convergence should be new. I didn't see it before.

Reproducibility: Yes

Additional Feedback:


Review 2

Summary and Contributions: The authors develop stochastic quasi-Newton methods for training deep neural networks (DNNs). In particular, they design Kronecker-factored block-diagonal type BFGS and L-BFGS methods. In numerical examples, they demonstrate the acceleration effect of the proposed methods for autoencoder feed-forward neural network models with either nine or thirteen layers applied to three datasets.

Strengths: 1. The authors develop new BFGS and L-BFGS methods using the feed-forward structure DNN of an inverse Hessian approximation. 2. They provide the proof of convergence of a stochastic Kronecker-factored quasi-Newton method.

Weaknesses: The paper is clear.

Correctness: The claim is correct. The method describes the Kronecker structures of the gradient and Hessian based on a single point, and then extend it to the expectation structure in learning problem. In these formulations, they formulate the approximated Hessian operators by BFGS.

Clarity: The paper is well written, expect a typo in "Our contributions...New BGFS..." This should be "New BFGS''.

Relation to Prior Work: There is also other type of natural gradient based on L^2 Wasserstein, which can be useful. Li, et.al. Natural gradient via optimal transport, Informaton geometry, 2018. Wang. et. al. Information Newton's flow: second-order optimization method in probability space, 2020.

Reproducibility: Yes

Additional Feedback: 1. For the construction of Hessian matrix for neural parameters, what is the major difference between the proposed method with the classical BFGS methods? What is the most challenging part in current work, on the approximation of Hessian and on the formulation Hessian inverse in neural networks? 2. How does the current method connect with the Fisher-Rao natural gradient, a.k.a. Hessian preconditioned gradient flow in neural networks? I have read the author's response. It totally addresses my questions. I increase my score from 7 to 9.


Review 3

Summary and Contributions: This paper proposes a stochastic Quasi-Newton (QN) method based on BFGS updates that exploits the structure of feed-forward neural networks. The technique approximates the loss Hessian as a block diagonal matrix where each block represents a layer based on a Kronecker-factored approximation. As blocks of the estimated inverse Hessian are factored into two terms, the authors propose separate BFGS-like updates with different damping schemes to ensure that the approximated matrix is positive definite (and simultaneously limiting the decrease in its smallest eigenvalue). Moreover, under not too restrictive assumptions they prove convergence of the proposed algorithm (without the double damping scheme). Through numerical experiments they demonstrate that the algorithm has better or on-par convergence speed on training data as typical first-order methods and KFAC (a related QN algorithm for deep neural networks using the Kronecker approximation).

Strengths: Exploiting curvature information in deep neural network training has been of interest to researchers in the community due to its potential to significantly reduce training time and therefore the problem investigated by this paper is very relevant to the NeurIPS community. The theoretical grounding of the algorithm is sound. Building upon the well-known KFAC framework, this paper brings new ideas to the table in the form of novel BFGS updates and damping schemes to approximate blocks of the Hessian inverse and hence avoiding the costly matrix inversion step present in KFAC. Table 2 demonstrates the favorable scaling with input/output dimensions compared to KFAC (d_i^3, d_o^3 due to inversion versus d_i^2 and d_o). Many insights of the BFGS framework apply to the proposed algorithm, and the main theorem provides a valuable convergence guarantee in a highly non-convex setting.

Weaknesses: There are some weaknesses of QN methods applied to deep neural networks that also somewhat limit the applicability of the proposed algorithm. First, there are additional hyperparameters compared to first-order methods that need to be tuned beyond learning rate, namely damping terms (two for this algorithm), decay parameter for calculating moving-average to stabilize BFGS updates and the memory-parameter p for LBFGS. The authors merged the two damping terms into a single hyperparameter assuming some relation between them and performed sensitivity analysis, however a systematic way of tuning all these hyperparameters to a new application remains a bit challenging. Second, generalization performance of deep networks trained via second-order methods might lag behind first-order methods such as small batch SGD and Adam, especially without careful hyperparameter tuning. The authors have chosen not to include generalization results in the experiments and argued that the focus of the paper is comparing optimization techniques. While I more or less agree with the authors, demonstrating that the proposed algorithm has comparable (or at least reasonable) generalization to the related techniques would better position it as a practical QN method. Lastly, in its current form the proposed algorithm is applied to neural networks with fully connected layers. In order for it to become more practical and applicable to current state-of-the-art architectures an efficient extension to convolutional layers and normalization layers with parameters would be important (such as KFAC for convolutional networks). Post rebuttal: I appreciate the authors' effort to answer my concerns. The test accuracy plots seem to verify that the proposed algorithm has good generalization performance. I updated my score accordingly.

Correctness: The claims and proofs presented in the main submission appear to be correct. The experimental validation and comparison to other algorithms is fair, however the network used for comparison is somewhat outdated (but it is understandable for comparison, the same network was used in the KFAC paper).

Clarity: The paper is well-written and clear for most parts. Initially, I found it a bit confusing that H is used for the inverse Hessian approximation, whereas it is standard to use the same notation for the Hessian. It might also be interesting to add more discussion on the structural approximation and its limitations and validity in the deep learning context, as it was somewhat lacking. There are a couple of typos, please check lines 63, 111, 194.

Relation to Prior Work: Authors clearly differentiate their work from the closely related KFAC algorithm throughout the paper. The literature review provides a good list of approaches to incorporate second-order information in the context of neural network optimization, however the authors could put more emphasis on highlighting how their algorithm is different from each of these algorithms in the Introduction. The contributions of this work are clearly stated in the paper.

Reproducibility: Yes

Additional Feedback: Some discussion on the large-scale application of the proposed algorithm, especially its distributed deployment on compute servers would be very interesting. Do the authors plan on looking into a distributed implementation of the algorithm?


Review 4

Summary and Contributions: In this paper, the authors propose efficient quasi-Newton methods for feedforward neural networks by exploiting the Kronecker-factored block-diagonal structure. The idea is built upon a deterministic version of BFGS method on this structure. Several damping schemes are adopted to address this positive-definite issue in BFGS.

Strengths: This work shows that the BFGS idea can be used while preserving the Kronecker-factored block-diagonal structure. Several damping techniques and variance reduction techniques (such as moving average) are also suggested in a mini-batch setting.  This work looks very interesting and new.

Weaknesses: In this work, the mini-batch size is m=1000, which seems to be too big. I wonder whether the proposed methods can be used in a small mini-batch setting (e.g. m=50 or m=100). The proposed methods mainly follow the key idea of the deterministic BFGS method. It is likely that the proposed methods work well since mini-batch gradients are not too noisy. This issue should be clarified.

Correctness: The algorithmic detail seems to be correct since it mainly follows the deterministic version of BGFS. I do not carefully check the convergence analysis. 

Clarity: I think this paper is well-written. It will be more clear if the authors explicitly mention the reason why two forward-backward passes are required in Algorithm 2 since the deterministic BFGS method only requires one forward-backward pass.  I think the reason is H_g and H_a should be updated over the same mini-batch, which requires us to compute a new pair (x_g,y_g) over the same mini-batch.

Relation to Prior Work: I am not aware of prior works in exploiting the Kronecker-factored block-diagonal for BFGS methods.

Reproducibility: Yes

Additional Feedback: It will be great to include plots using test loss since the focus of this paper is on practical algorithms for DNN. The authors also should report the performance of the proposed methods when the mini-batch size is reasonably small. I have read the author's response. It addresses my questions. I increase my score accordingly.

[Author Response · NeurIPS 2020]

We thank all the reviewers for their time and for raising several interesting questions. We also appreciate that the reviewers carefully read the paper, catching typos, and making useful suggestions. Please see our responses below.

**Reviewer #1**: The three minor issues raised by the reviewer will be addressed in the revision.

**Reviewer #2**: In response to [8.Additional feedback], the most significant difference between the proposed methods and a standard BFGS approach is that our approach requires much less storage and work per iteration (see Sec 1 and Tables 1,2). Also, our BFGS updates depend on variables that are computed in the course of computing the stochastic gradients, rather than the gradients themselves. The most challenging part of our work involves the development of our BFGS updates so that they provide good approximations to the inverse of a Kronecker-factored block-diagonal approximation of the Hessian matrix. The relationship of our methods to the Fisher-Rao natural gradient (NG) method is that we approximate the Hessian instead of the Fisher matrix. As in the KFAC method, we use a block-diagonal Kronecker product approximation, where the two sub-blocks in the Kronecker product of each diagonal block is further approximated using BFGS-based updating. We will update the description of our methods in our revision to better explain the connections and differences between them and Fisher-Rao and other BFGS methods. We thank the reviewer for alerting us to the papers on Wasserstein-based NG methods and will include them in the introduction.

**Reviewer #3**: The generalization performance of our algorithms is demonstrated in the following figures, which we will add to the paper. The upper (lower) row plots the training loss (testing error - i.e., mean squared error on the test set), respectively. Hyper-parameters (HPs) are set as in Fig 1 of the paper.

Regarding additional HPs, our experiments show that the key HPs for our methods are learning rate and damping constant, as is the case for KFAC and the adaptive gradient methods. Our methods are relatively insensitive to the other HPs, which can be set to default values. We will also highlight how our algorithms are different from related prior work and add more discussion on the structural approximations made and their limitations and validity in a DL context. Please see the response to Reviewer #2.

Our K-BFGS can be extended to other architectures, such as CNNs. Due to both time and space limitations, we focused only on fully-connected NNs, but our preliminary studies indicate that our approach works equally well on CNNs and we are currently working on this extension. We follow standard QN notation, where $B$ denotes the Hessian and $H$ its inverse. A distributed version of the proposed method would definitely be of interest and we are looking into this.

**Reviewer #4**: We have repeated our experiments using mini-batches of size 100 for all algorithms (see the figures below, where HPs are optimally tuned for batch sizes of 100) and our proposed methods continue to demonstrate advantageous performance, both in training and testing. These results show that our approach works as well for relatively small mini-batch sizes of 100, as those of size 1000, which are less noisy, and hence is robust in the stochastic setting employed to train DNNs.

The reviewer's understanding of the need for two forward-backward passes is correct. We will add this explanation to the paper. Please see the response to Reviewer #2 for our numerical results on testing error.

[Meta-Review · NeurIPS 2020]

This paper develops a quasi-newton optimization algorithm for training fully connected neural networks. The authors develop an LBFGS like approximation of the Hessian via a block kronecker product factorization. On the experimental side they demonstrate the acceleration effect of the proposed methods for autoencoder feed-forward neural network models with either nine or thirteen layers applied to three datasets. The authors also provide some theoretical guarantees on convergence to stationarity with additional assumptions. All reviewers found the paper interesting. Some reviewers raised concerns about the additional tuning parameters, lack of results for test error, and some limitations in the experiments (no result on classification or more complex data sets as numerical experiments use MNIST). Most of these concerns were alleviated based on the authors’ response and in particular the generalization error results provided in the authors feedback. I concur with the reviewers and think the development of second order methods which speed up training of neural networks is interesting and well suited to Neurips. While there are some limitations in the numerical experiments I think this paper takes an important step in the right direction and therefore recommend acceptance. I do encourage the authors to follow the reviewers’ suggestions to further improve their final manuscript including the addition of the generalization error curves.